# Probable New Species of Bacteria of the Genus *Pseudomonas* Accelerates and Enhances the Destruction of Perfluorocarboxylic Acids

**DOI:** 10.3390/toxics12120930

**Published:** 2024-12-22

**Authors:** Sergey Chetverikov, Gaisar Hkudaigulov, Danil Sharipov, Sergey Starikov

**Affiliations:** Ufa Institute of Biology, Subdivision of the Ufa Federal Research Centre of the Russian Academy of Sciences, 450054 Ufa, Russia; bio-logos@yandex.com (G.H.); male886@yandex.ru (D.S.); senik0406@gmail.com (S.S.)

**Keywords:** perfluorocarboxylic acids, biodegradation, association of bacteria, *Pseudomonas*, transformation, genome

## Abstract

Bacteria of the genus *Pseudomonas* are the most studied microorganisms that biodegrade persistent perfluoroorganic pollutants, and the research of their application for the remediation of environmental sites using biotechnological approaches remains relevant. The aim of this study was to investigate the ability of a known destructor of perfluorooctane sulfonic acid from the genus *Pseudomonas* to accelerate and enhance the destruction of long-chain perfluorocarboxylic acids (PFCAs), specifically perfluorooctanoic acid and perfluorononanoic acid, in water and soil in association with the strain *P*. *mosselii* 5(3), which has previously confirmed genetic potential for the degrading of PFCAs. The complete genome (5.86 million base pairs) of the strain 2,4-D, probably belonging to a new species of *Pseudomonas*, was sequenced, assembled, and analyzed. The genomes of both strains contain genes involved in the defluorination of fluorinated compounds, including haloacetate dehalogenase H-1 (*dehH1*) and haloalkane dehalogenase (*dhaA*). The strain 2,4-D also has a multicomponent enzyme system consisting of a dioxygenase component, an electron carrier, and 2-halobenzoate 1,2-dioxygenase (CbdA) with a preference for fluorides. The strain 2,4-D was able to defluorinate PFCAs in an aqueous cultivation system within 7 days, using them as the sole source of carbon and energy and converting them to perfluorheptanoic acid. It assisted strain 5(3) to convert PFCAs to perfluoropentanoic acid, accelerating the process by 24 h. In a model experiment for the bioaugmentation of microorganisms in artificially contaminated soil, the degradation of PFCAs by the association of pseudomonads also occurred faster and deeper than by the individual strains, achieving a degree of biodestruction of 75% over 60 days, with the perfluoropentanoic acid as the main metabolite. These results are of great importance for the development of methods for the biological recultivation of fluorinated organic pollutants for environmental protection and for understanding the fundamental mechanisms of bacterial interactions with these compounds.

## 1. Introduction

Halogenated chemicals are continuously entering various environmental objects due to industrial pollution and persist there for decades [1]. Perfluorocarboxylic acids (PFCAs) are industrial chemicals that, due to the characteristics of their chemical structure and their high chemical stability and resistance, have been widely used for over 50 years. The widespread use of PFCAs, particularly as fire-fighting foams in the military and mining sectors, along with their recently identified toxicity, highlights the scale of soil and groundwater contamination by these substances worldwide [2,3,4].

Alongside the well-known perfluorooctanoic acid (PFOA), perfluorononanoic acid (PFNA) can also be highlighted. The area of application for PFNA includes its use as a polymerization additive in the production of fluoropolymers [5] in the commercial product Surflon S-111 (CAS 72968-3-88, 74% by weight). PFNAs are very stable and do not degrade in the environment due to oxidative processes because of the strength of the carbon–fluorine bond. Like the eight-carbon PFOA, the nine-carbon PFNA is a toxic substance for the development of an organism and immune system [6] and it can disrupt the human thyroid hormone system, with possible negative consequences for pregnancy and the subsequent development of the fetus and child [7]. Their toxicity has been repeatedly confirmed for both animal and human health [8,9]. They not only move with water [10,11] but also accumulate in soils [12,13], thus entering cultivated food crops [14,15]. Due to their combined toxic properties concerning living organisms, they have been included in Annex B of the Stockholm Convention on Persistent Organic Pollutants [16].

Accordingly, the issues related to their degradation are relevant and require solutions. Initially, various physical (pyrolysis, membrane and adsorption separation, sonolysis) [17,18,19] and chemical methods (ozonation, plasma, photocatalytic and electrochemical reactions) [20,21,22] were used for this purpose (degradation, defluorination, destruction of PFCAs). However, considering their high costs for energy supply, multistage pre-treatment, and the formation of toxic byproducts [23], microbial destruction, primarily bacterial, is a relevant alternative. Biocatalytic technologies for the degradation of perfluoroorganic compounds are promising, environmentally friendly, and economical in terms of energy use. However, two to three decades ago, PFCAs were referred to as ‘forever chemicals’, and the inability of microorganisms to degrade them was attributed to a lack of time for their evolution and the absence of natural analogs to stimulate the evolution of the necessary enzymes [24]. But perhaps the time has come, and it is precisely ‘here and now’ that progress has been made in the cleavage of robust chemical bonds in persistent halogenated organic compounds, leading to their transformation into harmless substances. Initially, microorganisms learned to break carbon covalent bonds that do not contain fluorides (for example, CS, CO) during the biodegradation of perfluorocarboxylic (sulfonic) acids [25,26,27,28]. Not stopping there, they progressed to the effective cleavage of C-F bonds, thereby reducing the chemical stability of the organic molecule of perfluoroalkyl compounds and making it available for further degradation [29]. It also expanded the range of perfluoroalkyl compounds used as substrates by microorganisms to C_7_–C_10_, the defluorination of which has been confirmed through the detection of released fluoride ions [30].

In recent years, it has been shown that bacteria and their communities are capable of metabolizing certain PFCAs and forming a range of their metabolites [31,32]. One of the proposed pathways for the microbial destruction of PFCAs involves their oxidation to perfluorohexanoic acid, which has been demonstrated for the strain *Gordonia* sp. NB4-1Y [33]. Prior to this, several strains of bacteria from the genus *Pseudomonas* with a similar activity were known [34,35,36].

Recent reviews have touched upon the limited published materials regarding the enzymes and molecular tools of microorganisms involved in the destruction and transformation of PFCA through the release of fluoride ions and the protection against its toxic effects [37,38]. So far, the number of microorganisms and enzymes demonstrating such activity remains limited. A list has been presented, including dehalogenases of halogenated acids, reductive dehalogenases, laccases, desulfonases, and mechanisms of microbial resistance to intracellular fluoride [37,38].

Enzymatic biocatalysts for breaking C-F bonds under mild conditions have been discovered relatively recently, and they generally do not require the use of cofactors [39]. For example, several related dehalogenases with defluorination potential were found in the genome of the strain *Delftia acidovaerens* D4B, isolated from soil contaminated with PFCA, which were later identified as DeHa2 (haloacid dehalogenase) and DeHa4 (fluoroacetate dehalogenase) [40,41]. One of the latest reports in this cluster of studies concerned the discovery of the *rdhA* gene, which plays a key role in the defluorination of PFCAs by the strain *Acidimicrobium* sp. A6 [42].

We previously demonstrated that the strain *Pseudomonas mosselii* 5(3) is capable of both genetically and effectively degrading C_7_–C_10_ PFCAs to C_6_ [30], while the strain *Pseudomonas* sp. 2,4-D also showed biodestructive activity toward perfluorooctane sulfonate, breaking it down to perfluoroenanthic acid [28]. Given the high potential for a possible additive effect in the destruction of PFCAs through the associative use of these strains, this study involved the sequencing of the genome of a representative new species of the genus *Pseudomonas* strain 2,4-D, examining its genetic features and biodegradation potential to increase the degree and rate of PFCA destruction when cultivated in an aqueous environment and in artificially contaminated soil through the bioaugmentation of microorganisms.

## 2. Materials and Methods

### 2.1. Bacterial Strains and Conditions for Their Cultivation in Liquid Medium

Strain 5(3), identified by whole-genome sequencing as *Pseudomonas mosselii* and capable of degrading C_7_–C_10_ PFCAs [30], was isolated from pesticide-contaminated soil in the Yanaul district of the Republic of Bashkortostan of the Russian Federation. The project of its genome has been deposited in GenBank under the number JAUHUJ000000000 and the BioSample and BioProject numbers SAMN36271085 and PRJNA990579, respectively.

Strain 2,4-D was isolated from a soil sample taken from a facility producing halogenated organic compounds (Ufa, Republic of Bashkortostan), previously identified as *P. plecoglossicida* 2,4-D and described as a destructor of perfluorooctane sulfonic acid [28].

Strains 2,4-D and 5(3) have been deposited in the Microorganism Collection of the Ufa Institute of Biology, Subdivision of the Ufa Federal Research Centre of the Russian Academy of Sciences, under numbers UIB-53 and UIB-251.

The strains were cultivated at 28 °C in liquid Raymond mineral medium (g per liter of distilled water): NH_4_NO_3_–2.0, MgSO_4_ × 7H_2_O–0.2, KH_2_PO_4_–2.0, Na_2_HPO_4_–3, CaCl_2_ × 6H_2_O–0.01, Na_2_CO_3_–0.1 [43], with PFOA or PFNA (250 mg/L) as the sole source of carbon and energy in an orbital shaker-incubator ES-20/60 (SIA BIOSAN, Riga, Latvia) at 180 rpm.

To obtain the inoculum, the strain was cultivated on Raymond’s mineral medium with the addition of peptone (1 g/L) for 24 h. The biomass was harvested and washed with sterile distilled water before inoculation into experimental flasks. The resulting pellet was resuspended in mineral medium and added to the experimental flasks until an OD_600_ of 0.1 was reached.

For the estimation of bacterial quantity and the isolation of individual colonies, LB medium [44] was used, consisting of (g per liter of distilled water) tryptone—10, yeast extract—5, NaCl—5, and agar—15 g (PanReac, Barcelona, Spain).

To assess the effectiveness of PFCA degradation, the bacterium was cultured at 28 °C in a liquid mineral medium with individual C_8_–C_9_ PFCAs for 7 days. All experiments were conducted in three independent biological replicates.

### 2.2. Chemicals and Reagents

Perfluorocarboxylic acids—perfluorononanoic acid and perfluorooctanoic acid (both of high purity > 98%)—were purchased from Sigma Aldrich (St. Louis, MO, USA). Acetonitrile and methanol (HPLC grade) were acquired from Merck (Darmstadt, Germany). All other reagents were of analytical grade.

### 2.3. Sequencing, Assembly, and Annotation of the Genome

The genomic DNA of strain 2,4-D was extracted from biomass using the phenol-chloroform method [45]. Sequencing was performed using DNBSEQ-G50 equipment (MGI, Shenzhen, China), and paired-end reads of 100 bp were obtained.

Quality control of the reads was performed using ‘HTQC’ [46]. Low-quality (Q < 25), short (<100 bp) reads and adapter sequences were removed using Trimomatic version 0.39 [47]. Genomes were assembled using SPAdes software version 3.15.4 [48]. Error correction was carried out using Pilon version 1.23 [49] and Bowtie2 version 2.3.5.1 [50]. To confirm the presence of the assembled circular replicon, we analyzed the presence of overlapping ends. For strain identification, average nucleotide identity (ANI) parameters (https://www.ezbiocloud.net/tools/ani (accessed on 23 October 2024), [51]) and digital DNA-DNA hybridization (DDH) (https://ggdc.dsmz.de/ggdc.php (accessed on 23 October 2024), [52]) were used with default settings.

Contigs shorter than 500 bp were removed. Annotation was performed using Prokka version 1.14.5 [53]. Analysis of target gene sequences was conducted using the NCBI BLAST service [54], utilizing the nr/nt and WGS databases.

The FASTME 2.1.6.1 program [55] was used to construct the minimum evolutionary tree based on the obtained intergenomic distances. The tree was rooted in the middle [56] and visualized using https://itol.embl.de/ (accessed 2 October 2024) [57]. Species and subspecies were grouped according to [58,59], respectively. The OrthoANI algorithm [60] was used to calculate the average nucleotide identity (ANI) between strains and related strains.

### 2.4. Model Experiment on the Bioaugmentation of Strains 2,4-D and 5(3) and Their Association in Soil Artificially Contaminated with PFCAs

For the model experiment, we used chernozem soil collected from an organic farming enterprise (Republic of Bashkortostan, Russia) that was not contaminated with chemical organic compounds. The experiment included four lines, each treated with PFOA or PFOS to a final concentration of 10 mg/kg of dry soil: (1) soil (control); (2) soil augmented with the bacterial culture of strain 2,4-D (final culture concentration 2 × 10^6^ CFU/g of soil) (2,4-D); (3) soil augmented with the bacterial culture of strain 5(3) (final culture concentration 2 × 10^6^ CFU/g of soil) (5(3)); (4) soil augmented with an association of bacterial cultures of strains 2,4-D and 5(3) (final culture concentration 2 × 10^6^ CFU/g of soil). The cultivation was conducted at 28 ± 1 °C, and the soil moisture was maintained at 40% of the soil’s moisture capacity. The number of microorganisms was assessed using the dilution method on Raymond’s medium with the corresponding PFCA as a sole carbon source. The duration of the experiment was 60 days. Extraction of PFASs from soil samples was carried out using a modified method from Rankin et al. [61]. The prepared sample (3.5 g of dry soil) was transferred into polypropylene centrifuge tubes, previously washed with methanol. For PFAS extraction, a mixture of acetonitrile and water in a 9:1 ratio (8.5 mL) was used, with the addition of 400 µL of a 2 M NaOH solution. After adding the reagents, the samples were subjected to ultrasonic treatment in an ice bath for 15 min, followed by shaking (using a shaker) for 30 min and centrifugation for 10 min at 11,800× *g*. The resulting supernatants from the three extraction steps were combined in glass tubes. For sample preparation for analysis, the combined supernatant was evaporated to dryness under a stream of air. The dry residue was dissolved in two consecutive washes of acetonitrile and water in a 2:3 ratio, each with 0.5 mL. To assess the bacterial count, samples were inoculated onto Raymond’s medium with the corresponding PFCAs every 10 days of the experiment. All experiments were conducted in three independent biological replicates.

The efficiency of PFCA destruction was calculated using the formula
D(%) = 100 − ((C_t_ × 100)/C_0_),
where D is the destruction efficiency (%), C_t_ is the concentration of PFCA after a certain period of time, and C_0_ is the concentration of PFCA at the initial moment in time.

### 2.5. Isolation and Identification of the Biotransformation Products of PFCA

The content of PFCA in the medium was assessed and the products of its biotransformation were identified using a liquid tandem chromatography-mass spectrometer LCMS-IT-TOF (“Shimadzu”, Japan) at the Shared Equipment Center “AGIDEL” of the Ufa Scientific Center of the Russian Academy of Sciences, in ultrafiltrates (≤3 kDa) of culture media obtained by ultrafiltration using the “Vivaflow 50” system (Sartorius AG, Göttingen, Germany), as described in [28].

For chromatographic separation, a Shim-pack XR-ODS column (75 mm × 2.0 mm id, 2.2 μm) (Shimadzu, Kyoto, Japan) was used with a mobile phase consisting of 5 mM ammonium acetate in acetonitrile (A) and 0.1% acetic acid (*v*/*v*) in water (B). Elution was performed with a linear gradient as follows: from 0 min to 10 min, from 60% to 30% (B); from 10 to 20 min, from 30% to 60% (B). The separation was conducted at a flow rate of 0.25 mL/min, with an injection volume of 5 μL. Mass spectra were obtained using electrospray ionization (ESI) in the negative ion mode with the following parameters: high-voltage probe: −3.5 kV; nebulizer gas flow: 1.5 L/min; CDL temperature: 200 °C; heating block temperature: 50 °C; drying gas pressure: 150 kPa; TOF detector voltage: 1.57 kV. A trifluoroacetic acid solution was used as a standard sample for sensitivity and resolution calibration, as well as for mass calibration (ion trap and time-of-flight analyzer).

The structure of the obtained compounds was established based on a combination of mass spectrometry data, which focused on the fragmentation of the molecular ion, and software analysis using LCMS Solution version 3.60 (Shimadzu). The release of fluoride ions was determined using ion chromatography with an LC-20 Prominence HPLC system equipped with a CDD-10Avp conductivity detector (Shimadzu, Japan). The analyte separation was performed on a Shodex column (Shodex, New York, NY, USA) at a flow rate of 1 mL/min with an eluent composed of 1.8 mM Na_2_CO_3_ + 1.7 mM NaHCO_3_, using the Xenoic^®^ XAMS ASUREX-A100 suppressor (Diduco AB, Umeå, Sweden). A fluoride stock solution (1000 mg/L) was prepared from 2.21 g of NaF dissolved in 1L of deionized water. Fluoride calibration solutions in the range of 1 to 100 mg/L were prepared by serial dilutions of the stock solution with water.

The metabolic pathways were drawn using ChemDraw Ultra v. 12.0.2.1076.

### 2.6. Statistical Processing and Data Analysis

The statistical analysis was performed using Microsoft Office Excel 2021. All experiments were conducted in triplicate. Standard methods of parametric statistics were used to describe the research results: the arithmetic mean and standard deviation were calculated. The comparison of two groups was carried out using a two-tailed Student’s *t*-test. The critical level of significance was set at 0.05 for this study.

## 3. Results and Discussion

### 3.1. Identification of the Strain 2,4-D and Functional Annotation of Its Genome

Previously, based on analyses of the physiological and biochemical properties and the 16S rRNA gene, the strain 2,4-D was identified as *Pseudomonas plecoglossicida* [28]. The same study demonstrated its ability to transform perfluorooctanesulfonic acid. In the current work, we present the results of sequencing and obtaining a reliable draft assembly of the genome of the pure culture, which were used for identification and subsequent functional analysis. Sequencing and complete assembly of the genome of the strain *Pseudomonas* sp. 2,4-D revealed a circular chromosomal replicon of 5,861,101 bp (GC content: 61.84%). Other assembly characteristics are presented in Figure 1A. The genome project for strain 2,4-D has been deposited in GenBank under the number JAUHUI000000000 and the BioSample and BioProject numbers SAMN36271038 and PRJNA990562, respectively. Based on whole-genome sequencing, an attempt was made to determine the phylogenetic relationship between the studied strain and phylogenetically related strains of the genus *Pseudomonas* (Figure 1B). Following the recommendations in [62] and using Average Nucleotide Identity (ANI) and Digital DNA–DNA Hybridization (DDH) parameters, a comparison of the genome of the studied strain and the genomes of closely related type strains *P. kurunegalensis* RW1P2, *P. putida* NBRC14164, and *P. monteilii* DSM14164 was conducted. The ANI values ranged from 89.34% to 89.83%, while the DDH values ranged from 56.40% to 60.30%. According to the criteria for describing new bacterial species, these values should not be lower than 95% and 70%, respectively, compared to the type strains of known species.

None of the parameters allow the studied strain to meet the species threshold for classification as a specific species, and it is likely that the strain *Pseudomonas* sp. 2,4-D represents a new species. As the functional annotation shows, the genome contains genes for all the metabolic pathways necessary for the existence of an autonomous culture (Figure 1C). Considering the lack of the detailed taxonomy of this genus, and despite the long history of study and extensive databases, the genus is annually supplemented with new species.

Using the KEGG metabolic pathway and network analysis programs, it was found that the genome of the strain 2,4-D encodes several genes potentially involved in the biodegradation and metabolism of xenobiotics. This category contains 34 genes. The genome of the 2,4-D strain was analyzed for genetic elements potentially associated with the degradation of PFCA. It contains, similar to the genome of the PFCA-degrading strain 5(3) [30], genes potentially involved in the defluorination of fluorinated compounds, such as haloacetate dehalogenase H-1 (*dehH1*), haloalkane dehalogenase (*dhaA*), and alkane sulfonate monooxygenase (SsuE). However, unlike strain 5(3), the genome of the 2,4-D strain does not contain a fluoride ion transporter (CrcB), but it does have a multicomponent enzyme system consisting of a dioxygenase component, an electron transport protein, and a 2-halobenzoate 1,2-dioxygenase large subunit (CbdA) with a preference for fluorides, which has not been previously mentioned in the context of PFCA.

The presence of genes for haloacetate dehalogenase H-1 (*dehH1*), haloalkane dehalogenase (*dhaA*), or similar ones is also characteristic of other PFCA-degrading organisms, particularly *Delftia acidovaerens* D4B [40]. The *Acidimicrobium* sp. A6 has been found to contain genes encoding a homolog of reductive dehalogenase (RdhA) and a homolog of fluoroacetate dehalogenase (FceA), as well as two putative genes for halogen acid dehalogenases (*dhl_1* and *dhl_2*) [42]. Molecular dynamics simulations have been performed, and the docking of the enzyme complex in the corresponding configuration for PFAS substrates has been investigated [42].

The absence of the fluoride ion transporter gene (CrcB) likely negatively impacts the viability of the culture during the transformation of PFOS and the possibility to overcome the C_7_ barrier due to the toxic effects of the released fluoride. This is also supported by the data indicating that effective defluorination requires a high tolerance of the microbial host to the elevated intracellular concentrations of F^−^ [63]. The discovery of the putative fluoride ion transporter, the gene of CrcB, in the A6 genome highlights a significant problem posed by the toxicity of fluoride, a byproduct of the cleavage of the C–F bond during metabolism. The lack of a rapid and efficient mechanism for fluoride export would render this recently developed metabolic pathway detrimental to the organism, ultimately leading to its exclusion from the population. As noted in [42], this underscores the need for a reliable enzymatic apparatus, potentially involving ATP-consuming processes, to catalyze the cleavage of C–F bonds and mitigate the toxic effects of very high fluoride levels [64].

A multicomponent enzyme system, manifesting hydroxylation, defluorination, and decarboxylation properties, consisted of a dioxygenase component, an electron carrier, and the 2-halobenzoate 1,2-dioxygenase large subunit (CbdA) originally from *P. cepacia* 2CBS [65]. CbdA catalyzed the double hydroxylation of 2-halobenzoates with the concomitant release of halide and carbon dioxide. For strain 2,4-D, this enzyme had broad substrate specificity and was likely involved in the transformation of PFCAs.

It is worth highlighting the presence in the strain 2,4-D of the alternative vanadium nitrogenase gene of VnfA. Their hosts are used in associations to enhance various biotechnological processes [66], for example, supplying microorganisms with nitrogen by converting it into ammonia under microaerobic conditions and low temperatures to sustain their vitality [67].

So, the identification of genes related to specific pathways in bacterial genomes gives a prediction about their engagement in the processes. However, due to the complex regulation of gene expression, the actual activity of the enzymes involved in the defluorination and biodegradation processes of ‘forever chemicals’ still requires confirmation [68].

### 3.2. Biodegradation and Defluorination of PFCA

Both studied *Pseudomonas* strains, as previously shown [28,30], are capable grown in mineral media with perfluorinated organic acids as the sole source of energy and carbon, with a maximum microbial count in the culture liquid on the 7th day, completely transforming the substrate. The degree of the transformation of the substrate varied: strain 5(3) degraded them to C_6_ PFCA, while strain 2,4-D degraded them to perfluoroenanthic acid. Accordingly, an association of these microorganisms was formed to overcome this barrier. Associations or consortia of microorganisms are generally more effective than their individual counterparts, as they consist of different strains with broad and complementary enzymatic capabilities. Earlier, it was shown for persistent organohalogen substances such as polychlorinated biphenyl [69] and chlorobenzene [70].

In the current study, individual strains required 7 days for the complete degradation of PFCAs, but in association, they accomplished this 24 h faster with both PFOA (Figure 2A) and PFNA (Figure 2C). This was confirmed by the release of fluoride ions into the medium (up to 30 mg/L), and the amount of fluoride released in the case of the microbial association exceeded that of the individual strains by 1.6 to 2.2 times.

Considering that PFCAs are inert anionic substances, they are analyzed using liquid chromatography/tandem mass spectrometry (Figure 3A) (MS spectra shown in Appendix A); the detection of fluoride ions released during the biodegradation process is carried out using conductometric HPLC (Figure 3B).

The conducted LCMS analysis also showed a difference in the degree of transformation of PFCAs by microorganisms and their association (Figure 2B,D). At the end of cultivation in a medium containing PFOA (Figure 2B), for strain 2,4-D, we observed dissociated acid ions (*m*/*z* 363) of PFHpA (97% from the total amount of ions), for strain 5(3)—mostly PFHxA (*m*/*z* 313; 50%), and for their association—mostly PFPeA (*m*/*z* 263; 88%). Similarly, the degradation of PFNA (Figure 2D) occurred under the influence of bacteria and their association and resulted in the ion content of 97%, 81%, and 85%, accordingly. The obtained products differ in mass by the amount of CF_2_—groups cleaved by bacterial dehalogenases, i.e., by *m*/*z* 50.

Such bacterial activity of the biodegradation of perfluorinated substances, confirmed through the mineralization up to fluorine and probably with a presence in the bacterial genome of the genes responsible for that, has not been reported by authors in the literature.

To date, the closest result achieved involved the use of the bacterial strain *Acidimicrobium* sp. A6 for the removal of PFOA and PFOS in aerobic conditions, where the degree of biodegradation did not exceed 63% for a concentration of 100 mg/L over 100 days [19]. The removal of PFOA and PFOS has also been documented in systems with microbial consortia, showing a 16% reduction under the action of chemoorganotrophic bacteria and a 36% reduction under the action of a consortium of yeast and mold over 28 days. Their LC/MS analysis showed the production of monofluorinated fatty acids [31]. Additionally, data on the degradation of various perfluoroalkyl substances by a native consortium of microorganisms consisting of *Flexilinea floccule*, *Bacteriovorax stoplpii*, and *Sphingomonas* sp. are presented in [71], which detected PFOA, PFHxA, and PFHpA as metabolites in aqueous solutions after 10 months of incubation.

For the defluorination of PFCA, bacteria need to transport it into the cell and produce an active enzyme that catalyzes the cleavage of the C−F bond. We do not have information about the transport systems for PFCA import into the bacterial cell. On the other hand, we have data on the extracellular biodegradation of PFCAs. This capability of the studied strains is also supported by partially published data [72] regarding the extraction from their culture supernatants, partial purification, and molecular characterization of extracellular dehalogenases (with molecular weights of 72 and 80 kDa) that vary in structure and possibly in mechanisms of action.

Considering that strain 5(3) specializes in the degradation of PFCAs, while strain 2,4-D focuses on the destruction of PFOS, we conclude that the latter enhances the degree and speed of PFCA degradation. The genetically determined presence of a multienzyme system with broad substrate specificity, exhibiting decarboxylation, hydroxylation, and defluorination properties (the presence of the CbdA coding gene), increases the possibility of this conclusion.

There is growing interest in oxygenase enzymes and their potential ability to transform PFCAs through the catalysis of reactions with their functional groups [29]. The reason for this interest in oxygenases is their ability to catalyze the oxidation of various substances different from their primary substrates [73]. Similarly, the enzyme complex that was found in the strain 2,4-D, including dioxygenase, may potentially catalyze not only the destruction of chlorinated aromatic compounds but also perfluorinated compounds.

Additionally, the overcoming of the physiological barrier of fluoride toxicity in these bacteria during the degradation of PFCA is explained by the presence of a fluoride ion transporter (CrcB), which allows them to bypass this barrier and conduct defluorination at a new level in association.

### 3.3. Model Experiment on the Bioaugmentation of PFCA-Contaminated Soil

In the present study, model soil systems were developed to evaluate the potential application of a bacterial consortium as a bioremediation agent. It was demonstrated that the introduction of bacteria reduced the concentration of PFCA in the soil by 22–25% (strain 2,4-D) and 38–40% (strain 5(3)) over a period of 60 days (Figure 4A,D). The microbial titers in the soil increased linearly during this time, reaching values at the end of the experiment that differed from the initial levels by slightly more than an order of magnitude (Figure 4B,E). The use of the bacterial consortium resulted in a significant enhancement of PFCA degradation in the soil, achieving 75–78% over a period of 60 days, which was corroborated by improved microbial survival in the soil (the titer increased by two orders of magnitude over the course of the experiment). It should be noted that in model systems lacking the introduced microorganisms, there was virtually no degradation of PFCAs. According to the LCMS analysis of PFCAs and their degradation products (Figure 4C,F), the degradation of PFCAs in the soil differed in metabolite profiles from their biodegradation in aquatic environments. In all augmented soil experiments, a dissociated acid ion (with *m*/*z* 263) corresponding to perfluoropentanoic acid was detected. If the consortium was used, this dissociated acid ion was the predominant one, indicating an accelerated transformation of the initial PFCAs.

Bioremediation requires various microorganisms to break down and transform pollutants into energy, cellular biomass, and harmless byproducts. The effectiveness of bioremediation strategies can be enhanced through bioaugmentation—introducing specific biodegraders to accelerate the degradation of contaminants [74]. This method is also used in contexts involving halogenated organics [75,76] and PFAS [77].

The obtained data stand out positively against the backdrop of sparse information on this topic. Unfortunately, at the current moment, the authors have not found a significant number of scientific publications on the biodegradation of PFCAs in soils. This again refers to our earlier publication on the reduction of PFOS in soil by 75% over 3 months under the influence of bacteria of the genus Pseudomonas [28], without disclosing the metabolites of destruction. Additionally, in a series of enzymatic reactions of oxidative humification, it has been shown that PFOA can decompose in the presence of soybean meal (which contains high concentrations of natural organic mediators and multivalent metal ions) and laccases. The degree of destruction in the soil after 140 days was 40%. It was established that the degradation products of PFOA are partially fluorinated organic compounds. At the same time, the degradation mechanism was presumably initiated by the direct attacks of free radicals on C–C bonds, leading to a chain reaction of free radicals. Laccase enzymes also require a cofactor source and oxygen to maintain their ability to biotransform PFOA, which makes its application questionable in cases of real contamination [78]. The further degradation of PFCAs and their complete bacterial mineralization is likely a question that will be answered in the near future through continued investigation into the catabolic capabilities of the microorganisms studied.

In response to the question posed in reference [77]: “Rehabilitation of soils contaminated with poly- and perfluoroalkyl substances (PFAS): to mobilize, immobilize, or decompose?”, we provide a clear answer. Decomposition through microorganisms, even considering that the removal of PFASs in soil using bacteria will be complicated in practice, due to the complex mixture of PFAS compounds in the contaminated environment and limited biodegradation, is desirable.

And we will continue our research on the formation of the associations of microorganisms that have the ability to synthesize various enzymes and enzymatic systems for the destruction of toxic PFCAs, as well as studying the mechanisms of overcoming the toxic effects of fluoride. Another promising direction is the combined use of microorganisms and green plants, as well as various plant–microbe symbioses. For these purposes, we are selecting microorganisms capable of degrading PFCAs, which also have a number of beneficial properties for plants (plant growth stimulation).

## 4. Conclusions

The capabilities of a well-known destructor of perfluorooctanesulfonic acid from the genus *Pseudomonas* strain 2,4-D have been studied to accelerate and enhance the degree of destruction of long-chain PFCAs in association with *P. mosselii* 5(3)—the confirmed destructor of C_7_–C_10_ PFCAs. According to the data obtained from whole-genome sequencing, strain 2,4-D is likely classified as a new species within the genus *Pseudomonas*. The association of two *Pseudomonas* strains has the ability to decarboxylate and defluorinate PFCAs up to perfluoropentanoic acid in liquid medium and soil within 6 and 60 days, respectively, due to a gene complex that includes haloacetate dehalogenase H-1 (*dehH1*), haloalkane dehalogenase (*dhaA*), fluoride ion transporter (CrcB), and a multicomponent enzyme system consisting of a dioxygenase component, an electron transfer component, and the large subunit of 2-halobenzoate 1,2-dioxygenase (CbdA) with a high affinity for fluorides. The detection of the degradation and assessment of the transformation of anionic acids were carried out using LC-MS analysis, while the release of indicators for the mineralization of fluoride ions was determined by ion chromatography. Thus, it has been demonstrated that bacterial strains of the genus *Pseudomonas* in association with each other are capable of achieving their genetic potential for the biodegradation of PFCAs with an additive effect.

## Figures and Tables

**Figure 1 toxics-12-00930-f001:**
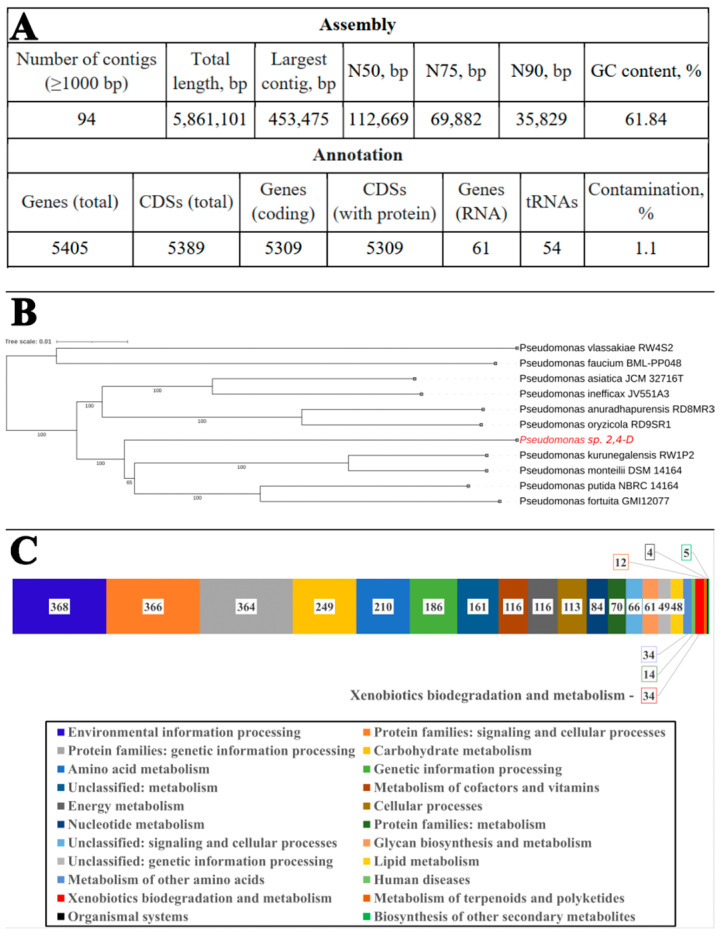
Assembly metrics of the strain 2,4-D (**A**): whole-genome phylogenetic tree within the genus *Pseudomonas* (**B**) and the number of genes associated with common functional categories in its genome according to KEGG classification (**C**).

**Figure 2 toxics-12-00930-f002:**
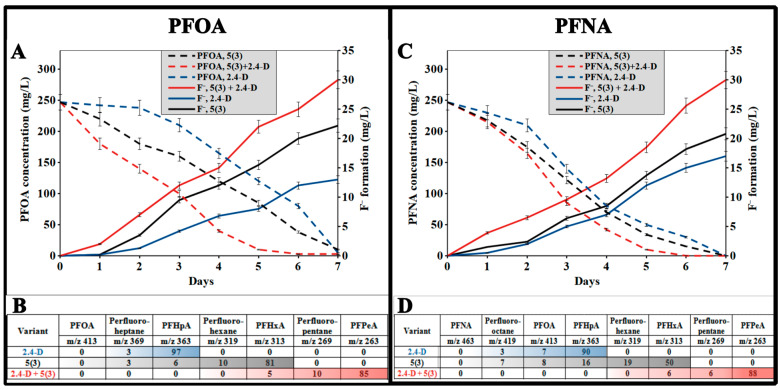
Dynamics of fluoride ion release and changes in PFCA concentrations during cultivation in liquid mineral medium ((**A**) with PFOA, (**C**) with PFNA) of *Pseudomonas* sp. 2,4-D strains (blue), *P. mosselii* 5(3) (black), their association (red), and degree of biodegradation potential based on component ratios (%) in the mixture on day 7 in the variants (**B**) with PFOA and (**D**) with PFNA. The data are presented as mean values. The error bars represent the s.d. from *n* = 3 replicate experiments.

**Figure 3 toxics-12-00930-f003:**
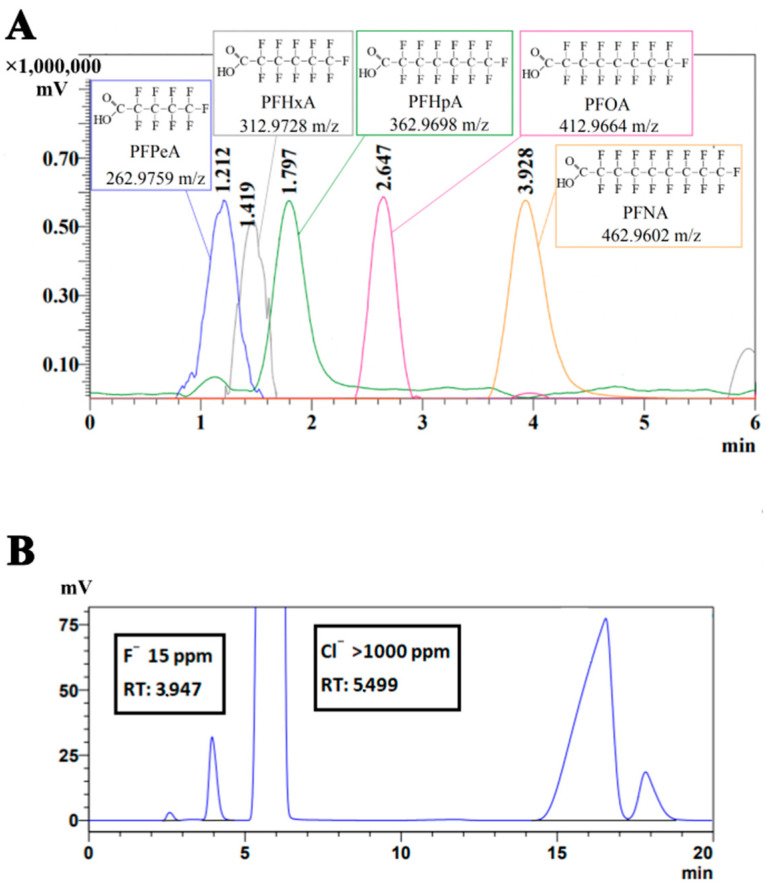
Temporal characteristics and chromatograms of PFCA separation on a chromatograph-mass spectrometer (**A**) and the detection of released fluoride ions by ion chromatography (**B**).

**Figure 4 toxics-12-00930-f004:**
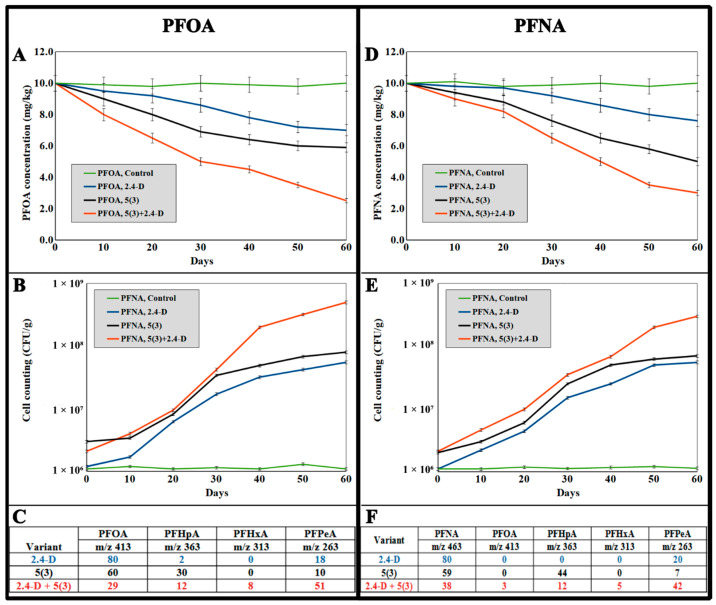
Kinetics of the growth of *Pseudomonas* sp. 2,4-D (blue), *P. mosselii* 5(3) (black), and the consortium of both (red) during pot assays of the biodegradation of PFOA (**A**,**B**) and PFNA (**D**,**E**) and the ratio (%) of the end products of PFOA (**C**) and PFNA (**F**) biodegradation (model experiment in soil). The data are presented as mean values. The error bars represent the s.d. from *n* = 3 replicate experiments.

## Data Availability

The project of the genome has been deposited in GenBank under the number JAUHUJ000000000 and the BioSample and BioProject numbers SAMN36271085 and PRJNA990579, respectively.

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
