# Peer review of "Probable New Species of Bacteria of the Genus Pseudomonas Accelerates and Enhances the Destruction of Perfluorocarboxylic Acids"

_toxics, 2024, doi:10.3390/toxics12120930_

Round 1

Reviewer 1 Report

Comments and Suggestions for Authors

The manuscript explores an important and timely subject, namely the biodegradation of persistent perfluorocarboxylic acids (PFCAs) by Pseudomonas strains. The experimental design is robust, with comprehensive genomic analysis and clear documentation of the results. The manuscript presents valuable data, demonstrating the potential of microbial bioaugmentation in the remediation of PFCA contamination in water and soil.

However, there are a few areas where the manuscript could be improved to enhance clarity, structure, and depth of analysis.

Specific Comments:

1. Abstract and Introduction:

   - The abstract provides a clear summary of the study’s findings. However, it could be more concise, especially in terms of the detailed explanation of the experimental results.

   - The introduction effectively highlights the importance of PFCA degradation and the environmental implications. However, the comparison to previous research on similar bacterial strains could be more directly linked to the novelty of this study.

2. Materials and Methods:

   - The methods section is well-detailed, particularly with regard to the genome sequencing and PFCA degradation assays. However, more information on the controls used in the experiments would be helpful. Specifically, were there any experiments where no bacteria were added to the contaminated soil or water to assess background degradation rates?

3. Results and Discussion:

   - The results are presented in a logical order, but there is a need to further explain the biological relevance of certain findings, especially the role of the multi-component enzyme system in PFCA degradation. The discussion should elaborate more on how the identified genes contribute to the biotransformation process and how this compares with other bacterial strains reported in the literature.

   - The discussion section could benefit from a more thorough comparison with previous studies, especially those that have examined the bioremediation potential of other bacteria for PFCA degradation.

4. Figures and Tables:

   - The figures are generally clear, but Figure 1B, showing the phylogenetic tree, could benefit from a clearer label for strain 2,4-D. Furthermore, in Figure 2 and Figure 3, it would be helpful to include a brief description of the significance of the observed peaks and metabolites in the legend.

5. Conclusion:

   - The conclusion adequately summarizes the main findings but could be more explicit in stating the implications for practical applications, such as in environmental bioremediation or industrial uses. The manuscript would benefit from a more detailed discussion on the future directions of this research.

Author Response

The authors thank the reviewer for the work done to improve the quality of our article. We are ready to answer questions and accept comments.

  1. Abstract and Introduction:

- The abstract provides a clear summary of the study’s findings. However, it could be more concise, especially in terms of the detailed explanation of the experimental results.

Thank you. We have made it more compact and shorter.

- The introduction effectively highlights the importance of PFCA degradation and the environmental implications. However, the comparison to previous research on similar bacterial strains could be more directly linked to the novelty of this study.

The authors agree with the reviewer regarding the comparison with similar bacterial strains; however, the point is that, for now, we can only compare ourselves, for instance, in terms of the degradation of persistent pollutants in soil, primarily with our own data. Other than strain 2,4-D, there are no additional data in the literature. Nonetheless, we have made some additions.

  1. Materials and Methods:

- The methods section is well-detailed, particularly with regard to the genome sequencing and PFCA degradation assays. However, more information on the controls used in the experiments would be helpful. Specifically, were there any experiments where no bacteria were added to the contaminated soil or water to assess background degradation rates?

Yes, such control variants were included. Variants without the introduction of microorganisms were, of course, tested by us, but no degradation of the persistent pollutant was observed; these data are now presented in the figure and in the text.

  1. Results and Discussion:

- The results are presented in a logical order, but there is a need to further explain the biological relevance of certain findings, especially the role of the multi-component enzyme system in PFCA degradation. The discussion should elaborate more on how the identified genes contribute to the biotransformation process and how this compares with other bacterial strains reported in the literature.

The authors thank the reviewer for the comment. The detection of such a multi-component enzymatic system in the PFCA-degrading strain is itself a significant result. The presence of this system contributes to the degree of PFCA transformation, while the mechanisms of action remain not fully elucidated. We will continue to work on this aspect as well. It is challenging for us to expand the discussion regarding the comparison of the identified genes' roles in the biodegradation of PFCA relative to other bacterial strains. There is limited literature on this topic, and we consider ourselves somewhat pioneers in this area of research. We have made efforts to present the data as comprehensively as possible.

- The discussion section could benefit from a more thorough comparison with previous studies, especially those that have examined the bioremediation potential of other bacteria for PFCA degradation.

We have made efforts to present the data and discuss them as comprehensively as possible.

  1. Figures and Tables:

- The figures are generally clear, but Figure 1B, showing the phylogenetic tree, could benefit from a clearer label for strain 2,4-D. Furthermore, in Figure 2 and Figure 3, it would be helpful to include a brief description of the significance of the observed peaks and metabolites in the legend.

The position of the strain 2,4-D on the phylogenetic tree is highlighted in color (Figure 1), part of Figure 3 has been moved to the supplementary materials, and Figures 2 and 4 have been revised in accordance with the reviewers' suggestions.

  1. Conclusion:

- The conclusion adequately summarizes the main findings but could be more explicit in stating the implications for practical applications, such as in environmental bioremediation or industrial uses. The manuscript would benefit from a more detailed discussion on the future directions of this research.

The issue of perfluorocarbon substances has gained significant attention at high-level forums, culminating in discussions at the Stockholm Convention. Naturally, bioremediation methods will be essential in addressing this problem, and we believe that a combined approach utilizing comprehensive purification techniques alongside phytoremediation will be effective. Another promising avenue involves the synergistic application of microorganisms and green plants, exploring various plant-microbe symbioses. To this end, we are actively selecting microbial strains with beneficial properties for plants, particularly those that can stimulate growth. Regarding the use of microorganisms, we will continue our research on forming associations of microbes capable of synthesizing various enzymes for degradation and studying the mechanisms to overcome the toxic effects of fluoride ions.

Reviewer 2 Report

Comments and Suggestions for Authors

 The article submitted for peer-review addresses the important issue of removing persistent pollutants from the environment, among which halogenated organic compounds and their degradation products are particularly toxic.

General comments:

1. Drawings need to be improved to be more readable, this concerns the descriptions they contain, axis captions, etc.

2. English needs to be improved by a professional translator. By the way, please note where abbreviations should be in the plural and where singular. Currently, in many places this has not been taken into account (e.g. l. 109, 116, 271, 285, 288 …..and many more). Moreover, in many places the sentences are too long and complex, which makes it difficult to understand.

3. Please check carefully the correctness of all genes description which is usually in italics and with a lower case letter (e.g. l. 274 ssuE, l. 282 rdhA ….) and for expression products (usually proteins) without italics with a capital letter (e.g. l. 21, l. 277, l. 302 for CbdA, l. 275 for CrcB, ….) . In many places in the text it seems to me that these rules were not applied. Please check not only in the references to the work, but also for other bacterial models in published data.

4. After you enter an abbreviation into a manuscript for the first time, do not do it multiple times. Please use either the abbreviation or the full name.

5. The conclusions need to be rewritten to be much shorter and more concise, and the „Results” section renamed to "Results and discussion" in line with the content. In addition, the discussion section should contain more details.

6. The conclusions need to be rewritten to be much shorter and more concise.

More detail comments are mentions below:

Abstract and Introduction

l. 17 This statement is too categorical; please change it similarly as in the title

l. 82 and l. 93 Please, add a citation to these statements.

Materials and methods

l. 127 In the description of Raymond mineral medium there are no units for the listed ingredients. Please correct it.

l. 188 Since the unit of rotational speed (rpm) depends on the size and type of rotor, please specify the speed in g.

l. 190 Are You sure that it was air and not nitrogen? Using air could oxidize material.

l. 219 Please, add a citation/-s to the statement. Was a standard used to determine the amount of F?

Results

l. 281 Please add genus name Acidimicrobium sp. to be more clear and Ref. on the end of sentence.

l. 306-310 This part is not clear, it is not clear what refers to current research and what to cited literature. It needs to be rewritten. And the last sentence should be the first in this paragraph in my opinion.

l. 311-315 as above

e.g. Besides, the identification of genes related to specific pathway in bacterial genomes give prediction about their engagement in the processes. However, due to complex regulation .....[ref.]

l. 317 In my opinion, instead ‘actively grew’ should be ‘can ….”

l. 319-320 I propose to remove the fragment „reaching ….. this time.”

l. 320, l. 336 I suppose You thought „different degree” (?)/ degree instead depth

l. 325 Please, add a citation/-s to the statement.

l. 333 Please, add a scale on the left Y axis in the figures and on the right adding (10 x). I also suggest adding titles in A and C as PFOA and PFNA, respectively. The figure should be a whole so that there is no need to refer to the text. Therefore, they should also include a legend for the graphs, and not necessarily in the caption.

l. 339 What does „special non-traditional properties” mean? Please expand on this so there is no doubt.

l. 345 (Figure 3A) I would propose to move spectra of PFCAs included in A into Supplementary materials.

l. 349 In my opinion instead Fig. 2B,D it should be separate Table placed after this acapit and removed from Fig. 2.

l. 355 It is not clear to me, on what data the Authors based this statement?

l. 356 „biodestruction”(?) ot ‘biodegradation’, Morover, this statement is too categorical („confirmed” (?)). Check earlier comment for l. 311.

l. 366 Consortium containing of which strains? Please, list tchem.

l. 369-374 It is not clear for me. Authors write both about the transport system of F into bacterial cells and wright after it about „culture supernatants” (?).Conciseness does not have to mean disinformation.

l. 379 Next statement which is too categorical („supports”).

l. 397 Please add after what period?

l. 401- 406 After the word „analysis” please add of what? Moreover, this part should first contain a description of the obtained results and only then a comparison with literature data. It requires editing and supplementation. In addition, the phrase "predominant species" is imprecise.

l. 410 Please change the caption to be more clear. e.g.

Kinetics of growth Pseudomonas sp. 2,4-D, P. mosselii 5(3) and the consortium of both during pot assays of biodegradation of PFOA (A,B) and PFNA (D,E) and ratio (%) of end products of PFOA (C) and PFNA (F) biodegradation.

 And please, add legent in Figure for the strain description.

Comments on the Quality of English Language

English must be improved.

Author Response

The authors thank the reviewer for the work done to improve the quality of our article. We are ready to answer questions and accept comments.

General comments:

  1. Drawings need to be improved to be more readable, this concerns the descriptions they contain, axis captions, etc.

The figures have been significantly revised in accordance with the general and particular feedback from the reviewer.

  1. English needs to be improved by a professional translator. By the way, please note where abbreviations should be in the plural and where singular. Currently, in many places this has not been taken into account (e.g. l. 109, 116, 271, 285, 288 …..and many more). Moreover, in many places the sentences are too long and complex, which makes it difficult to understand.

The text in English was requested to be edited by Professor Kudoyarova, an expert in the English language and a frequent author of publications in various journals published by MDPI.

  1. Please check carefully the correctness of all genes description which is usually in italics and with a lower case letter (e.g. l. 274 ssuE, l. 282 rdhA ….) and for expression products (usually proteins) without italics with a capital letter (e.g. l. 21, l. 277, l. 302 for CbdA, l. 275 for CrcB, ….) . In many places in the text it seems to me that these rules were not applied. Please check not only in the references to the work, but also for other bacterial models in published data.

This is our lapse. We have corrected the errors

  1. After you enter an abbreviation into a manuscript for the first time, do not do it multiple times. Please use either the abbreviation or the full name.

It's our fault. Fixed.

  1. The conclusions need to be rewritten to be much shorter and more concise, and the „Results” section renamed to "Results and discussion" in line with the content. In addition, the discussion section should contain more details.

The conclusions have been shortened, the section 'Results' has been renamed to 'Results and Discussion', and the discussion has been expanded.

  1. The conclusions need to be rewritten to be much shorter and more concise.

The conclusions have become shorter and more concise.

More detail comments are mentions below:

Abstract and Introduction

  1. 17 This statement is too categorical; please change it similarly as in the title

The wording of the proposal has been changed.

  1. 82 and l. 93 Please, add a citation to these statements.

The link has been added.

Materials and methods

  1. 127 In the description of Raymond mineral medium there are no units for the listed ingredients. Please correct it.

Fixed it. A unit of measurement has been added.

  1. 188 Since the unit of rotational speed (rpm) depends on the size and type of rotor, please specify the speed in g.

Presented in g.

  1. 190 Are You sure that it was air and not nitrogen? Using air could oxidize material.

Yes, it was the air that was used. Oxidation is not the case with PFOA.

  1. 219 Please, add a citation/-s to the statement. Was a standard used to determine the amount of F?

The link has been added.

In order to prepare a fluoride stock solution of 1000 mg/L 2.21 g of salt NaF was dissolved in 1 L of deionized water. Fluoride calibration solutions in the range of 1 to 100.0 mg/L were prepared by serial dilutions of the stock solution with water.

Results

  1. 281 Please add genus name Acidimicrobium sp. to be more clear and Ref. on the end of sentence.

The ref. has been added.

  1. 306-310 This part is not clear, it is not clear what refers to current research and what to cited literature. It needs to be rewritten. And the last sentence should be the first in this paragraph in my opinion.

The text has been revised in accordance with the reviewers' suggestions.

  1. 311-315 as above

The text has been revised in accordance with the reviewers' suggestions.

  1. 317 In my opinion, instead ‘actively grew’ should be ‘can ….”

Done.

  1. 319-320 I propose to remove the fragment „reaching ….. this time.”

Corrected.

  1. 320, l. 336 I suppose You thought „different degree” (?)/ degree instead depth

Corrected.

  1. 325 Please, add a citation/-s to the statement.

citation/-s to the statement was made.

  1. 333 Please, add a scale on the left Y axis in the figures and on the right adding (10 x). I also suggest adding titles in A and C as PFOA and PFNA, respectively. The figure should be a whole so that there is no need to refer to the text. Therefore, they should also include a legend for the graphs, and not necessarily in the caption.

The text has been revised in accordance with the reviewers' suggestions.

  1. 339 What does „special non-traditional properties” mean? Please expand on this so there is no doubt.

PFCAs are inert anionic compounds, in contrast to most other persistent organic pollutants, and their analysis cannot be performed using commonly available HPLC; instead, a mass spectrometer is required. This is what was meant.

  1. 345 (Figure 3A) I would propose to move spectra of PFCAs included in A into Supplementary materials.

Thank you for your valuable comments, which helped us improve the presentation of the data in the figure. The spectra have been moved to the supplementary materials. Instead, to maintain the informative nature of the figure, m/z values extracted from the total ion current (TIC) are indicated in the figure.

  1. 349 In my opinion instead Fig. 2B,D it should be separate Table placed after this acapit and removed from Fig. 2.

We decided to keep it this way to maintain the compactness of the article.

  1. 355 It is not clear to me, on what data the Authors based this statement?

Using the example of the strain 2,4-D, which under the action of enzymes converts PFOA to PFHpA by cleaving a CF2 group (reducing the mass by m/z 50) - 1 cycle. PFNA is converted to PFHpA through PFOA by cleaving 2 CF2 groups, i.e., 2 cycles.

  1. 356 „biodestruction”(?) ot ‘biodegradation’, Morover, this statement is too categorical („confirmed” (?)). Check earlier comment for l. 311.

We agree with the reviewer. Changes have been made to the text.

  1. 366 Consortium containing of which strains? Please, list tchem.

We agree with the reviewer. Changes have been made to the text.

  1. 369-374 It is not clear for me. Authors write both about the transport system of F into bacterial cells and wright after it about „culture supernatants” (?).Conciseness does not have to mean disinformation.

Thank you for your comment. This point has been given special attention, and corresponding revisions have been made to the text of the article.

For the defluorination of PFCA, bacteria need to transport it into the cell and produce an active enzyme that catalyzes the cleavage of the C−F bond. We do not have information about transport systems for PFCA import into the bacterial cell. On the other hand, we have data on extracellurar biodegradation of PFCAs. This capabilities of the studied strains are also supported by partially published data  regarding the extraction from their culture supernatants, partial purification, and molecular characterization of extra-cellular dehalogenases that vary in structure and possibly in mechanisms of action.

  1. 379 Next statement which is too categorical („supports”).

We have revised the wording to: '...increases the likelihood of this conclusion.'

  1. 397 Please add after what period?

In 60 hours

  1. 401- 406 After the word „analysis” please add of what? Moreover, this part should first contain a description of the obtained results and only then a comparison with literature data. It requires editing and supplementation. In addition, the phrase "predominant species" is imprecise.

Annoying translation error, we have resolved it.

  1. 410 Please change the caption to be more clear. e.g.

Kinetics of growth Pseudomonas sp. 2,4-D, P. mosselii 5(3) and the consortium of both during pot assays of biodegradation of PFOA (A,B) and PFNA (D,E) and ratio (%) of end products of PFOA (C) and PFNA (F) biodegradation.

 And please, add legent in Figure for the strain description.

We agree with the reviewer's comment and have made the necessary changes.

Round 2

Reviewer 1 Report

Comments and Suggestions for Authors

The authors have thoroughly revised the manuscript based on the reviewer's comments and submitted the revised version along with a detailed response letter. Upon verification, the scientific content and technical details have been adequately addressed, the language is clear and fluent, and the structure is well-organized, meeting the publication standards of the journal. I recommend the manuscript for acceptance and publication.